# Direct Reward Fine-Tuning on Poses for Single Image to 3D Human in the Wild

**Seunguk Do, Minwoo Huh, Joonghyuk Shin, Jaesik Park**
Seoul National University

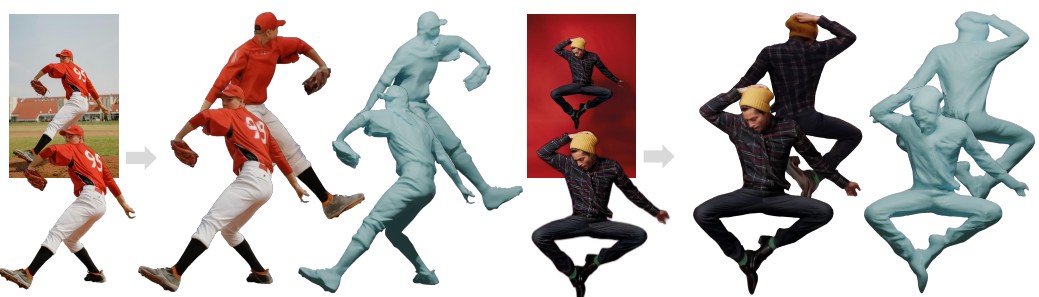

Figure 1: We propose DrPose (**D**irect **R**eward Fine-tuning on **Pose**s), a method to post-train a multi-view diffusion model for enhanced posture of reconstructed 3D humans in dynamic and acrobatic scenarios.

## ABSTRACT

Single-view 3D human reconstruction has achieved remarkable progress through the adoption of multi-view diffusion models, yet the recovered 3D humans often exhibit unnatural poses. This phenomenon becomes pronounced when reconstructing 3D humans with dynamic or challenging poses, which we attribute to the limited scale of available 3D human datasets with diverse poses. To address this limitation, we introduce DrPose, *Direct Reward fine-tuning algorithm on Poses*, which enables post-training of a multi-view diffusion model on diverse poses without requiring expensive 3D human assets. DrPose trains a model using only human poses paired with single-view images, employing direct reward fine-tuning to maximize PoseScore, which is our proposed differentiable reward that quantifies consistency between a generated multi-view latent image and a ground-truth human pose. This optimization is conducted on DrPose15K, a novel dataset that was constructed from an existing human motion dataset and a pose-conditioned video generative model. Constructed from abundant human pose sequence data, DrPose15K exhibits a broader pose distribution compared to existing 3D human datasets. We validate our approach by evaluating on conventional benchmark datasets, in-the-wild images, and a newly constructed benchmark, with a particular focus on challenging human poses. Our results demonstrate consistent qualitative and quantitative improvements across all benchmarks. Project page: https://seunguk-do.github.io/drpose.

## 1 INTRODUCTION

3D human models are essential assets across multiple industries, including visual media production (such as games and movies), product and industrial design, and e-commerce platforms for fashion. While multi-view scanning systems and manual design processes currently dominate 3D human crafting workflows, single-view 3D human reconstruction technology has garnered attention due to rapid technical advances and its practical advantages in scenarios where capturing multiple camera angles is either impractical or impossible.

Recent advances in this technology have been driven by the adoption of image-to-multi-view (I2MV) diffusion models, which have enhanced reconstruction quality for occluded body parts invisible in the input image (Pan et al., 2024; Peng et al., 2024; Li et al., 2024b; He et al., 2024; Xue et al., 2024; Ho et al., 2023a). This approach typically employs a two-stage pipeline: first generating multi-view images from a single input using a diffusion model, then lifting these views into 3D space through either implicit reconstruction (Saito et al., 2019; Ho et al., 2023a) or explicit reconstruction techniques (Li et al., 2024b; Palfinger, 2022; Xiu et al., 2022a). Compared to previous works, which directly reconstruct a 3D structure from the input-view feature (Saito et al., 2019; 2020) or works utilizing an estimated SMPL model (Xiu et al., 2022b;a), multi-view diffusion-based approaches have the benefit of using the powerful prior of diffusion models for the fine details on the unseen regions from the input-view.

Despite these advancements, a bottleneck persists that limits real-world applicability. Recovered 3D humans with a multi-view diffusion model often exhibit unnatural postures, especially when target poses are dynamic and challenging, such as extreme athletic movements or acrobatic postures. We argue that this limitation stems from the limited scale of publicly available training datasets (Yu et al., 2021; Han et al., 2023; Ho et al., 2023b) with diverse poses. This scarcity arises from the costs of recruiting diverse subjects and capturing them in varied poses using multi-view stereo setups, which are further compounded by privacy concerns that complicate the release of public data.

Our key insight for overcoming this challenge is to align an I2MV diffusion model with 3D human poses from an existing human motion dataset (Lin et al., 2023) that provides broader coverage of pose distributions than existing 3D human datasets. To this end, we first generate a single-view image for each pose using a pose-conditioned diffusion model (Men et al., 2025), constructing a dataset we call DrPose15K. We then post-train the I2MV diffusion model on this dataset using our proposed method DrPose, a *direct reward fine-tuning algorithm on poses*. DrPose maximizes PoseScore, our proposed differentiable reward function that quantifies the consistency between ground-truth 3D poses and the latent images generated by the I2MV model.

Our evaluation demonstrates that I2MV models post-trained with DrPose achieve improvements in single-view 3D human reconstruction quality both quantitatively and qualitatively. These improvements are consistent across all datasets, including conventional benchmarks (Yu et al., 2021; Ho et al., 2023b), in-the-wild images, and MixamoRP, our new evaluation benchmark designed to assess performance on complex and dynamic human poses.

Our key contributions are:

- We propose DrPose, a novel post-training algorithm for enhancing the alignment of an image-to-multi-view (I2MV) model with natural poses in dynamic and complex scenarios.
- We construct DrPose15K, a dataset comprising human poses from a motion dataset (Lin et al., 2023) paired with generated single-view images conditioned on each pose.
- Through quantitative evaluation, we demonstrate that our method achieves consistent improvements across all datasets, including conventional benchmarks and our proposed MixamoRP.

## 2 RELATED WORKS

### 2.1 SINGLE-VIEW 3D HUMAN RECONSTRUCTION

Single-view 3D human reconstruction remains a long-standing challenge in computer vision and graphics. Early approaches focused on recovering parametric human models (Loper et al., 2023; Pavlakos et al., 2019) but often lacked fine-grained details such as clothing and facial features (Bogo et al., 2016; Zhang et al., 2021; 2023a; Sun et al., 2021). A major advance was introduced by PIFu (Saito et al., 2019), which demonstrated that detailed 3D human shapes could be learned from a single image using implicit functions trained on 3D scan datasets. This inspired numerous extensions, including methods that (1) utilize normal maps to enhance surface quality (Saito et al., 2020; Xiu et al., 2022b;a), (2) utilize SMPL prior (Xiu et al., 2022b;a; Zhang et al., 2023b; Zhuang et al., 2025), (3) recover relightable textures (Alldieck et al., 2022), and (4) generate animation-ready avatars (Huang et al., 2020; He et al., 2021; Peng et al., 2024). Recently, generative models have further advanced the field by improving reconstruction quality for previously unseen views by

adopting *score distillation sampling* (Huang et al., 2023; Wang et al., 2025; AlBahar et al., 2023; Wang et al., 2024) or training a multi-view diffusion model (Pan et al., 2024; Peng et al., 2024; Li et al., 2024b; He et al., 2024; Xue et al., 2024; Hu et al., 2025). However, when these models receive images of out-of-distribution poses as input, their outputs exhibit unnatural postures. To address this, we propose a new approach that leverages motion data (Lin et al., 2023) to augment pose coverage and fine-tune multi-view diffusion models, thereby improving performance on diverse poses.

## 2.2 DIRECT REWARD FINE-TUNING OF DIFFUSION MODEL

Recent research has explored methods for post-training diffusion models to align them with human preferences better, building on the success of reinforcement learning techniques in large language models. This alignment process typically involves three key components: (1) starting with a pre-trained text-to-image diffusion model, (2) developing a reward model that evaluates attributes such as aesthetic quality, detail fidelity, and semantic alignment, and (3) optimizing the diffusion model to maximize these reward signals. Initial approaches utilized reinforcement learning (RL) objectives to maximize human preferences (Lee et al., 2023; Black et al., 2023; Fan et al., 2023). Building on human preference data, researchers have developed differentiable neural networks that can evaluate input images (Xu et al., 2023; Kirstain et al., 2023; Wu et al., 2023). Leveraging these advances, direct reward fine-tuning methods have recently emerged that post-train diffusion models using differentiable reward scores (Prabhudesai et al., 2024; Clark et al., 2023; Wu et al., 2024), thereby demonstrating faster convergence than RL-based approaches. In this work, we adopt DRTune (Wu et al., 2024), a state-of-the-art reward fine-tuning method, as the foundation for DrPose.

## 3 METHOD

This section describes our proposed method for aligning an image-to-multi-view (I2MV) diffusion model to natural postures in dynamic or complex cases, thereby enhancing the quality of its integrated single-view 3D human reconstruction pipeline. We first present DrPose, a novel post-training algorithm that aligns multi-view diffusion models to produce natural and accurate poses across diverse scenarios (Sec. 3.1). Next, we describe the construction of DrPose15K, our proposed training dataset with diverse pose coverage that DrPose operates on (Sec. 3.2). Finally, we describe the 3D reconstruction pipeline that uses an I2MV diffusion model post-trained via DrPose with explicit carving (Sec. 3.3).

### 3.1 DRPOSE

We introduce DrPose (Direct Reward Fine-tuning on Poses), a post-training algorithm for an I2MV diffusion model on a dataset, $D = \{I_i, \theta_i\}$, where $I_i, \theta_i$ are an input image and the ground-truth human pose. The core idea is to maximize the consistency between the generated multi-view latent images from $I_i$ and $\theta_i$, thereby better aligning the I2MV diffusion model with diverse poses in $D$ using PoseScore, our proposed differentiable reward.

---

**Algorithm 1** DrPose

**Dataset:** Image-pose pairs $D = \{I_i, \theta_i\}$
**Inputs:** I2MV diffusion model with initial weights $\omega_0$, reward model $r$, the number of training timesteps $K$, maximum early stop timestep $m$
Initialize $\omega = \omega_0$
**while** not converged **do**
    $s = \text{randint}(1, T - K \lfloor \frac{T}{K} \rfloor + 1)$
    $t_{\text{train}} = \{s + i \lfloor \frac{T}{K} \rfloor \mid i = 0, 1, \ldots, K - 1\}$
    $t_{\min} = \text{randint}(1, m)$
    $(I, \theta) \sim D$
    $\mathbf{x}_T \sim \mathcal{N}(0, \mathbf{I})$
    $\mathcal{L}_{\text{KL}} = 0$
    **for** $t = T, \cdots, 1$ **do**
        $\hat{\epsilon} = \epsilon_\omega(\texttt{stop\_grad}(\boldsymbol{x}_t), I, t)$
        **if** $t \notin t_{train}$ **then**
            $\hat{\epsilon} = \texttt{stop\_grad}(\hat{\epsilon})$
        **else**
            $\hat{\epsilon}_0 = \epsilon_{\omega_0}(\texttt{stop\_grad}(\boldsymbol{x}_t), I, t)$
            $\mathcal{L}_{\text{KL}} = \mathcal{L}_{\text{KL}} + \mathbb{E}(||\hat{\epsilon} - \hat{\epsilon}_0||)$
        $\hat{\boldsymbol{x}}_0 = (\mathbf{x}_t - \sigma_t \hat{\epsilon})/\alpha_t$
        **if** $t == t_{min}$ **then**
            $x_0 \approx \hat{x}_0$
            break
        $\boldsymbol{x}_{t-1} = \alpha_{t-1}\hat{\boldsymbol{x}}_0 + \sigma_{t-1}\hat{\epsilon}$
    $\mathcal{L}_{\text{reward}} = 1 - r(\hat{\boldsymbol{x}}_0, \theta)$
    $\omega \leftarrow \omega - \eta \nabla_\omega(\mathcal{L}_{\text{reward}} + w_{\text{KL}} \cdot \mathcal{L}_{\text{KL}})$

---

Building on DRTune (Wu et al., 2024), the previous reward fine-tuning algorithm, $x_0$ is generated from a noise $x_T \sim \mathcal{N}(0, \mathbf{I})$ while blocking the gradients of the denoising network input and sampling denoising steps for training, as described in Algo. 1. This strategy enables optimization of early denoising steps while maintaining computational efficiency. Then the reward loss $L_{\text{reward}} = 1 - r(\boldsymbol{x}_0, \theta)$ is computed using PoseScore, a differentiable reward function denoted as $r$.

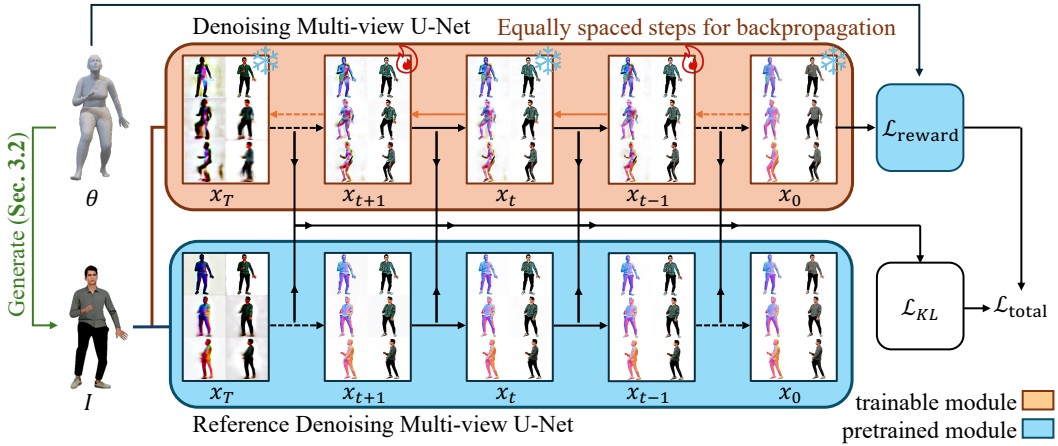

Figure 2: Illustration of DrPose presented in Algo 1. A Denoising multi-view U-Net $\epsilon_\omega$ is trained to minimize $\mathcal{L}_{\text{total}} = \mathcal{L}_{\text{reward}} + w_{\text{KL}} \cdot \mathcal{L}_{\text{KL}}$. Multi-view latent images $x_0$ are generated from $x_T \sim \mathcal{N}(0, \mathbf{I})$, and $\mathcal{L}_{\text{reward}}$ is computed from $x_0$ and the ground-truth 3D human pose $\theta$. For efficiency, only a subset of denoising steps is sampled for training. Concurrently, $\mathcal{L}_{\text{KL}}$ is computed as the KL divergence between $\epsilon_\omega$ and the frozen reference U-Net $\epsilon_{w_0}$ at intermediate denoising steps. For clarity, only 3 of the 6 multi-view images are shown.

**Differentiable Reward.** To quantify the consistency between a multi-view latent image $x_0$ and a GT pose $\theta$, we develop PoseScore, a differentiable reward model denoted as $r$. To compute consistency, it first projects both $x_0$ and $\theta$ onto the $\hat{I}_{\text{skel}}$ and $I_{\text{skel}}$ images, which depict the human skeletal structure. To convert $x_0$ into $\hat{I}_{\text{skel}}$, a U-Net based skeletal image predictor $g_{\text{skel}}$ is pretrained on the existing 3D human datasets (Ho et al., 2023b; Yu et al., 2021). Moreover, $I_{\text{skel}}$ can be drawn from the pose parameter $\theta$ by projecting the 3D human joints $J(\theta)$ onto the image planes corresponding to the viewpoints of the generated images. Then the reward is computed as follows:

$$r(\boldsymbol{x}_0, \theta) = -\mathbb{E}(||\hat{I}_{\text{skel}} - I_{\text{skel}}||) = -\mathbb{E}(||g_{\text{skel}}(\boldsymbol{x}_0) - \mathcal{R}(J(\theta))||), \tag{1}$$

where $\mathcal{R}$ is the rendering of the 3D human joints into the skeletal images into the viewpoints of $x_0$.

**KL divergence regularization.** As with previous direct reward fine-tuning methods (Wu et al., 2024; Prabhudesai et al., 2024), we find that training the denoising U-Net with PoseScore, a differentiable reward, leads to reward hacking, whereby image quality degrades while the reward score continues to increase. To address this, we augment the training objective with a KL divergence regularization term $L_{\text{KL}}$ alongside $L_{\text{reward}}$. This regularization computes $\mathbb{E}(||\hat{\epsilon} - \hat{\epsilon}_0||)$, where $\hat{\epsilon}$ represents the predicted noise from the trainable diffusion model at some timestep $t \in t_{\text{train}}$, and $\hat{\epsilon}_0$ is the corresponding prediction from the initial diffusion model. This constraint prevents the model's generated images from deviating excessively from its original results while optimizing for reward maximization.

To summarize, DrPose operates the steps in Algorithm 1 to minimize the following objective:

$$\min_{\omega} \mathbb{E}_{(I,\theta) \sim D} \left[ \mathcal{L}_{\text{reward}}(I, \theta) + w_{\text{KL}} \cdot \mathcal{L}_{\text{KL}}(I) \right]. \tag{2}$$

## 3.2 Construction of DrPose15K

We construct DrPose15K, a training dataset containing dynamic and challenging 3D human poses paired with single-view images, by leveraging Motion-X (Lin et al., 2023), a human motion dataset, and MIMO (Men et al., 2025), a pose-conditioned image-to-video(I2V) model, as illustrated in Figure 3. From the Motion-X dataset, we utilize the AIST (Li et al., 2021) subset due to its comprehensive coverage of diverse pose distributions. To reduce redundancy among the 300K available poses, we apply farthest-point sampling to select 1.5K poses. Then, we add the 9 temporal neighbors for each selected pose to create a pose sequence for input to the MIMO, yielding a total of 15K

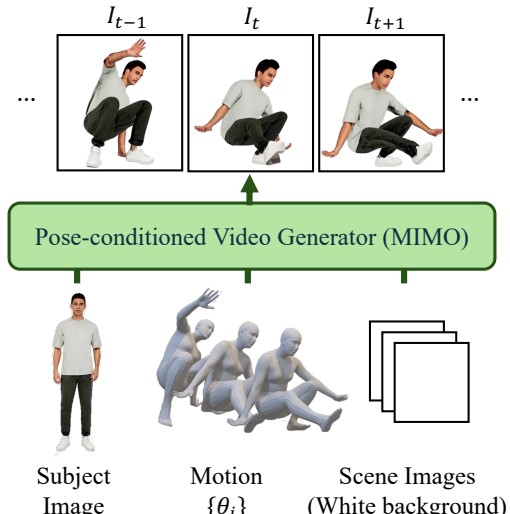

Figure 3: Construction process for DrPose15K. We employ a pose-conditioned video generator model (Men et al., 2025) to generate single-view images from 3D human poses.

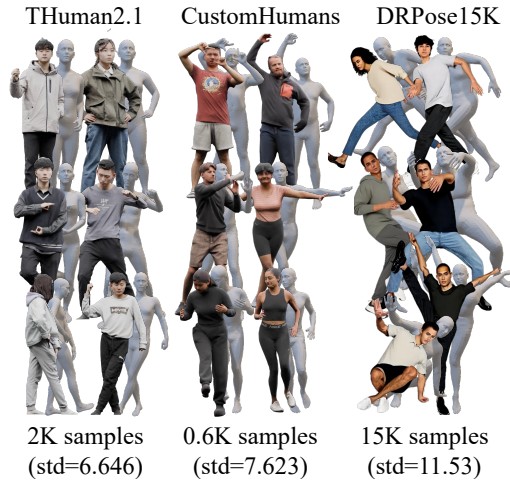

Figure 4: Comparison of pose diversity between conventional 3D human datasets (Yu et al., 2021; Ho et al., 2023b) and our proposed Dr-Pose15K. Our dataset has a higher standard deviation of SMPL-X joint locations than other datasets.

poses. Finally, we use MIMO to animate full-body human images from Generated Photos according to these pose sequences, generating corresponding single-view images for each 3D pose in our dataset.

To quantitatively assess the pose diversity of DrPose15K compared to conventional 3D human datasets (Ho et al., 2023b; Yu et al., 2021), we compute the standard deviation of SMPL-X joint positions across each dataset, focusing exclusively on the 22 body joints while excluding facial and hand joints. Note that for conventional datasets, we include both training and test splits in this analysis. As shown in Figure 4, DrPose15K exhibits a **1.73×** larger standard deviation compared to THuman2.1 (Yu et al., 2021). Moreover, with 14.7K poses, compared to 647 in CustomHumans and 2,445 in THuman2.1, DrPose15K provides broader coverage of pose distributions.

### 3.3 3D HUMAN RECONSTRUCTION WITH EXPLICIT CARVING

We show the overview of the pipeline for reconstruction of 3D humans from a single-view image in Fig. 5. In this pipeline, we employ an explicit carving to fit the generated multi-view images into 3D humans, following Li et al. (2024b). The pipeline first generates both normal maps and RGB images from the input image using a multi-view diffusion model post-trained with our DrPose. 3D human mesh recovery then proceeds through three sequential steps: SMPL-X initialization, differentiable remeshing (Palfinger, 2022), and appearance fusion. This approach delivers superior geometric detail compared to methods using pretrained implicit networks (Ho et al., 2023a; Pan et al., 2024)

## 4 EXPERIMENTS

### 4.1 IMPLEMENTATION DETAILS

**DrPose** During the post-training, we employ the DDIM sampler with $T = 20$ total denoising steps and $K = 2$ training steps. We set the maximum early-stopping timestep to $m = 8$ and weight the KL divergence loss at $w_{KL} = 0.01$. For computing $\mathcal{L}_{KL}$, we use mean squared error to estimate $||\hat{\epsilon} - \hat{\epsilon}_0||$.

**Denoising U-Net** To evaluate our proposed method DrPose, we apply it to two image-to-multi-view (I2MV) diffusion models, Era3D (Li et al., 2024a) and PSHuman (Li et al., 2024b), respec-

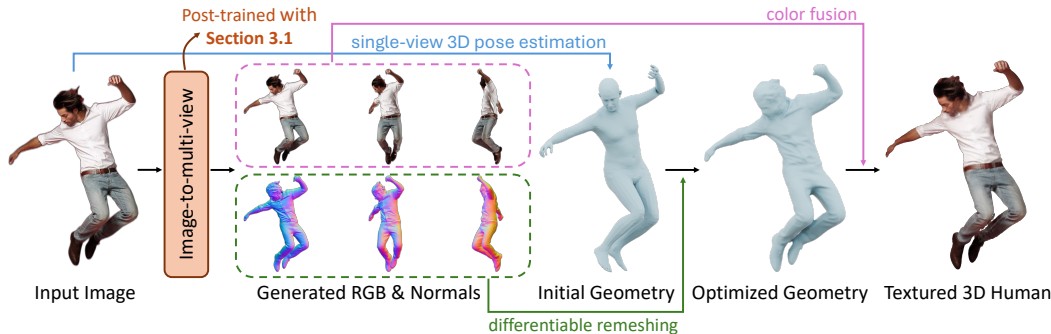

Figure 5: Overview of our 3D human reconstruction pipeline. In this pipeline, the multi-view normal and RGB images are generated from the input image using an image-to-multi-view (I2MV) diffusion model. These images are then converted into a 3D representation using explicit human carving (Li et al., 2024b). In this work, we propose post-training the I2MV diffusion model to achieve better alignment with accurate poses in dynamic and acrobatic scenarios. For clarity, only 3 of the 6 multi-view images are displayed for normal maps and RGB images.

tively, following their original architectures and initializing the weights $\omega_0$ accordingly. The model is fine-tuned on a single NVIDIA H200 GPU using a batch size of 2 with gradient accumulation over 2 steps for 18K iterations.

**Differentiable Reward** For computing the reward, we use binary cross entropy loss and LPIPS to estimate $||\hat{I}_{\text{skel}} - I_{\text{skel}}||$. The skeletal images $\hat{I}_{\text{skel}}$ and $I_{\text{skel}}$ both have 23 channels, with each channel corresponding to one joint. We use THuman2.1 (Yu et al., 2021) and the training subset of CustomHumans (Ho et al., 2023b) as our training datasets, comprising approximately 3K scans. To get six-view normal and color images, we render the 3D scans using Blender's Cycles engine (Community, 2018) with an orthographic camera configuration. The reward model is trained on four NVIDIA RTX 6000 Ada GPUs with a batch size of 16 over 10K iterations.

## 4.2 SINGLE-VIEW 3D HUMAN RECONSTRUCTION

### 4.2.1 BASELINES

We compare our approach against single-view 3D human reconstruction methods guided by SMPL (Xiu et al., 2022a; Ho et al., 2023a), as well as multi-view diffusion-based methods (Wu et al., 2023; Li et al., 2024a;b).

- **ECON** (Xiu et al., 2022a) estimates front and back depth maps using an estimated SMPL-X prior, then fuses these depth maps for a complete 3D human body. It does not support texture reconstruction and trains its depth estimation network on 500 scans from THuman2.0 (Yu et al., 2021).

- **SiTH** (Ho et al., 2023a) generates 512×512 px. RGB images for front and back views using an estimated SMPL-X prior, subsequently converting them to 3D via an SDF network. The diffusion model is trained on THuman2.0.

- **H3D(Human3Diffusion)** (Xue et al., 2024) produces four 256×256 px. RGB multi-view images, which are then converted to 3D using a 3DGS reconstruction network. The multi-view diffusion model is trained on 6K human scans, combining both public datasets and commercial datasets.

- **Era3D** (Li et al., 2024a) generates six 512×512 px. normal and RGB images using a diffusion network trained on Objaverse (Deitke et al., 2023). For fair comparison, we fine-tune this model on 3K scans from THuman2.1 and CustomHumans (Ho et al., 2023b) datasets.

- **PSHuman** (Li et al., 2024b) produces six 768×768 px. normal and RGB images using a diffusion network trained on THuman2.1 and CustomHumans datasets.

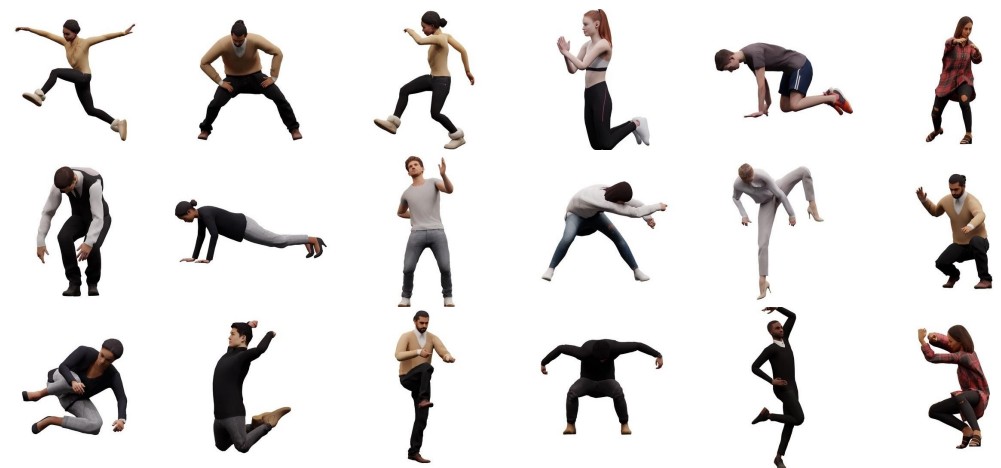

Figure 6: Representative visualizations for the MixamoRP benchmark. The 18 meshes shown were randomly sampled from the complete dataset containing 60 meshes.

### 4.2.2 BENCHMARKS

All baseline models in 4.2.1 and ours are evaluated quantitatively in the following three benchmarks:

- **THuman2.1-test** contains 60 human scans selected from the full THuman2.1 (Yu et al., 2021) dataset. The split follows Li et al. (2024b).

- **CustomHumans-test** contains 60 human scans selected from the full CustomHumans dataset, which consists of 600 human scans. The split follows Ho et al. (2023a).

- **MixamoRP** is our proposed benchmark containing 60 human scans, constructed by assigning 60 distinct poses collected from Mixamo animation, to 15 different Renderpeople 3D models, with 4 poses per model(see Fig. 6 for its visualization).

Test splits from CustomHumans (Ho et al., 2023b) and THuman2.1 (Yu et al., 2021) are commonly used benchmarks for evaluating single-view 3D human reconstruction methods. While these benchmarks include dynamic poses such as dancing or jumping, they lack extremely challenging configurations (see Figure 4) such as breakdancing or bat swinging.

**Construction of MixamoRP**    To establish more rigorous evaluation criteria for 3D human reconstruction under extreme pose variations, we introduce MixamoRP, a novel benchmark specifically designed to assess reconstruction performance on challenging pose configurations. To this end, we animate commercial rigged 3D models from Renderpeople (Renderpeople, 2023) using Mixamo (Inc., 2025) animations. Table 5 specifies character names, animations, descriptions, and frame indices for reproducibility, and Fig. 6 presents representative visualizations of MixamoRP.

### 4.2.3 EVALUATION PROTOCOL

We evaluate all baselines using their default configurations, with both Era3D and PSHuman variants employing 40 denoising steps and a classifier-free guidance (Ho & Salimans, 2022) scale of 3.0.

For each mesh scan, we render input images from 3 evenly distributed azimuthal views, yielding 180 input views per benchmark. To evaluate geometric accuracy, we report three metrics in Table 1: Chamfer Distance (CD), Normal Consistency (NC), and F-Score. To compute the Chamfer Distance, we uniformly sample 100K points per mesh.

For appearance evaluation, we report three metrics in Table 2: PSNR, SSIM, and LPIPS. To compute these metrics, we render images of both the prediction and the ground truth from 6 evenly distributed azimuthal views distinct from the input views.

Table 1: Quantitative comparisons of geometry quality on single-view human reconstruction benchmarks. Our proposed benchmark MixamoRP is described in Section 4.2.2. Era3D* represents the original Era3D model fine-tuned on CustomHumans and THuman2.1 training splits using conventional DDPM loss. Ours (Era3D) denotes the Era3D model post-trained with our proposed DrPose on DrPose15K.

| Method | THuman2.1-test | | | CustomHumans-test | | | MixamoRP | | |
|---|---|---|---|---|---|---|---|---|---|
| | CD↓ | NC↑ | f-Score↑ | CD↓ | NC↑ | f-Score↑ | CD↓ | NC↑ | f-Score↑ |
| ECON | 101.6465 | 0.6311 | 8.5244 | 126.1430 | 0.6205 | 6.4700 | 166.5384 | 0.5705 | 5.2173 |
| SiTH | 63.3041 | 0.6790 | 14.9221 | 71.9378 | 0.6713 | 12.7957 | 158.2729 | 0.5685 | 6.5176 |
| H3D | 75.8328 | 0.5959 | 12.2189 | 94.0864 | 0.5872 | 10.5563 | 149.2832 | 0.5400 | 7.2219 |
| Era3D* | 55.4071 | 0.6976 | 16.2928 | 63.1260 | 0.6914 | 14.1348 | 150.0118 | 0.5916 | 7.3487 |
| PSHuman | 52.9643 | 0.7194 | 18.6308 | 52.2187 | 0.7272 | 18.5624 | 137.2814 | 0.5876 | 8.2065 |
| Ours (Era3D) | 41.1770 | 0.7265 | 20.7102 | 44.3811 | 0.7310 | 20.1153 | 126.0622 | 0.5887 | 8.3071 |
| Ours | 42.0529 | 0.7252 | 19.6811 | 44.1326 | 0.7336 | 18.9600 | 126.5312 | 0.5998 | 8.8185 |

Table 2: Quantitative evaluation of 3D human reconstruction quality. Six RGB views evenly distributed in azimuth are rendered to compute appearance metrics. Our proposed benchmark MixamoRP is described in Section 4.2.2. Era3D* represents the original Era3D model fine-tuned on CustomHumans and THuman2.1 training splits using conventional DDPM loss. Ours (Era3D) denotes the Era3D model post-trained with our proposed DrPose on DrPose15K.

| Method | THuman2.1-test | | | CustomHumans-test | | | MixamoRP | | |
|---|---|---|---|---|---|---|---|---|---|
| | PSNR↑ | SSIM↑ | LPIPS↓ | PSNR↑ | SSIM↑ | LPIPS↓ | PSNR↑ | SSIM↑ | LPIPS↓ |
| SiTH | 17.7473 | 0.8829 | 0.1371 | 18.7462 | 0.8599 | 0.1533 | 15.6096 | 0.8118 | 0.2182 |
| H3D | 17.3290 | 0.8656 | 0.1561 | 17.6244 | 0.8328 | 0.1882 | 15.2037 | 0.7915 | 0.2440 |
| Era3D* | 17.7662 | 0.8826 | 0.1410 | 18.6322 | 0.8581 | 0.1584 | 17.5337 | 0.8623 | 0.1519 |
| PSHuman | 18.3880 | 0.8922 | 0.1286 | 18.9082 | 0.8612 | 0.1538 | 17.5931 | 0.8638 | 0.1499 |
| Ours (Era3D) | 20.3563 | 0.9050 | 0.1134 | 18.9286 | 0.8644 | 0.1508 | 17.5064 | 0.8662 | 0.1474 |
| Ours | 20.8594 | 0.9078 | 0.1063 | 19.1887 | 0.8638 | 0.1476 | 17.6632 | 0.8646 | 0.1465 |

### 4.2.4   RESULTS

As Table 1 and Table 2 present, our results demonstrate that DrPose consistently improves reconstruction quality of the base model across all benchmarks. We attribute this to our proposed DrPose's ability to enhance the accuracy of reconstructed posture on diverse poses, as seen in Figure 7, 8 and 9.

**Ablation Study on the base model**   We conduct an ablation study on the base model by post-training Era3D* using DrPose. As reported in Table 1 and Table 2, the Era3D-based model shows similar performance across all benchmarks. However, because the PSHuman-based model qualitatively yields better results on face regions, we chose PSHuman as our base model.

In Figure 10 and Table 3, we analyze the trained $g_{skel}$ of PoseScore introduced in Section 3.1. The evaluation is conducted on SMPL and scan mesh pairs from the test splits of CustomHumans and THuman2.1. The scan meshes are rendered as multi-view normal and color images, which are then encoded as latent images using PSHuman's VAE. These latent images are fed into $g_{skel}$ to produce skeletal images. Both the quantitative metrics and qualitative results demonstrate that $g_{skel}$ is sufficiently reliable to serve as a measure of consistency between latent images and poses.

Table 3: **Quantitative evaluation of $g_{skel}$ in PoseScore**

| Benchmark | PSNR | SSIM | LPIPS |
|---|---|---|---|
| THuman2.1-test | 22.4807 | 0.9337 | 0.0580 |
| CustomHumans-test | 24.4081 | 0.9536 | 0.0430 |

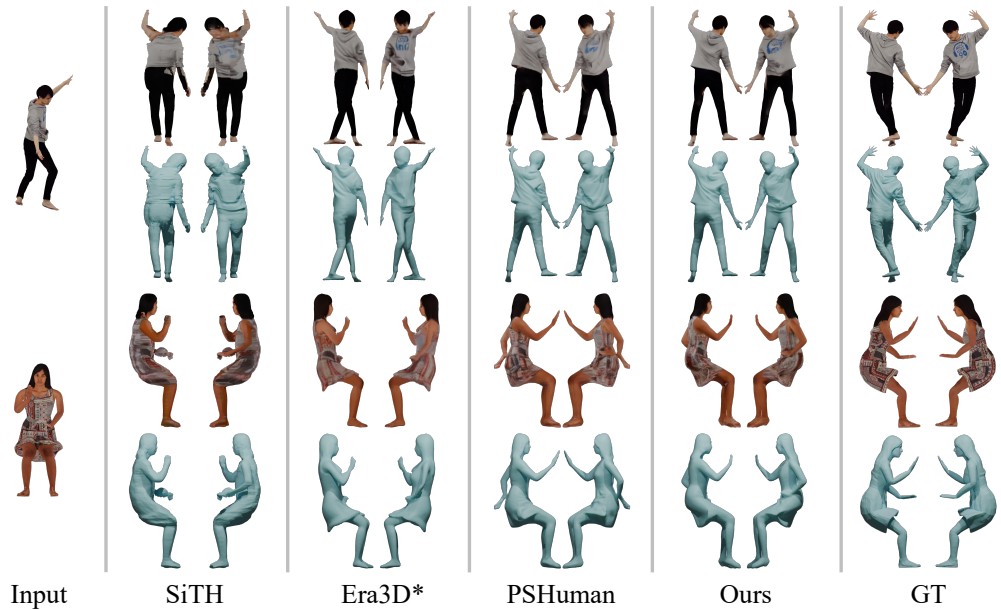

Input     SiTH     Era3D*     PSHuman     Ours     GT

Figure 7: Qualitative evaluation on the CustomHumans dataset. Era3D* denotes Era3D fine-tuned on CustomHumans and THuman2.1 datasets.

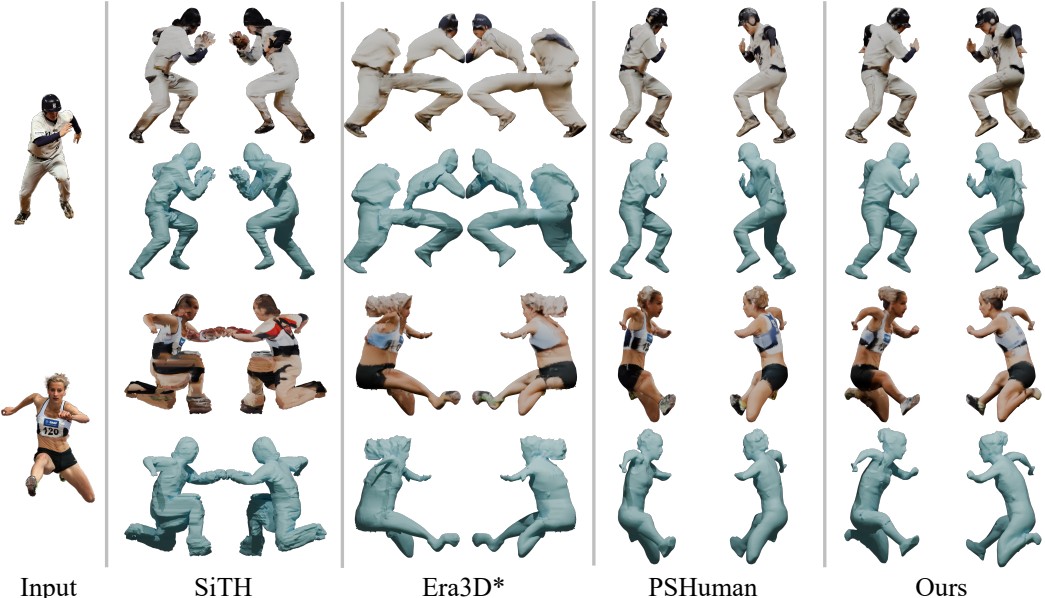

Input     SiTH     Era3D*     PSHuman     Ours

Figure 8: Qualitative evaluation on the internet-source images. Era3D* denotes Era3D fine-tuned on CustomHumans and THuman2.1 datasets.

## 5 CONCLUSION

We propose a novel approach to improve the pose accuracy of 3D humans reconstructed by multi-view diffusion models. Our method comprises three key contributions: (1) DrPose15K, a dataset featuring diverse poses with corresponding single-view images, (2) DrPose, an algorithm that enables post-training of multi-view diffusion models on this dataset; and (3) MixamoRP, a benchmark for evaluating reconstruction under challenging poses. Our post-trained model shows consistent quality improvements across all benchmarks. We discuss the ethics and reproducibility statements in Appendix A.6 and A.7.

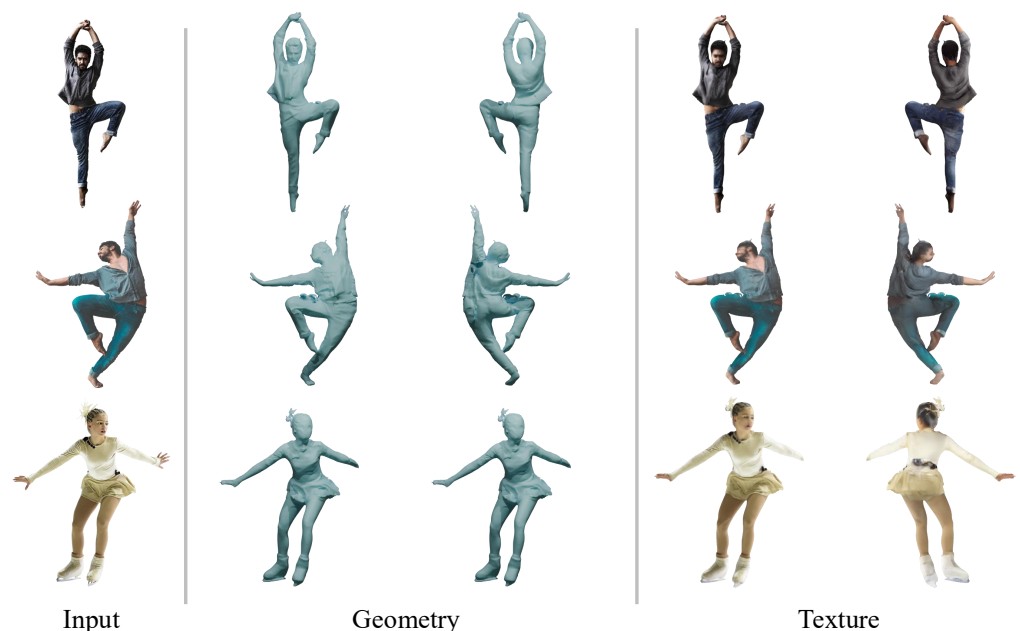

Input       Geometry       Texture

Figure 9: Qualitative results of DrPose on in-the-wild internet images.

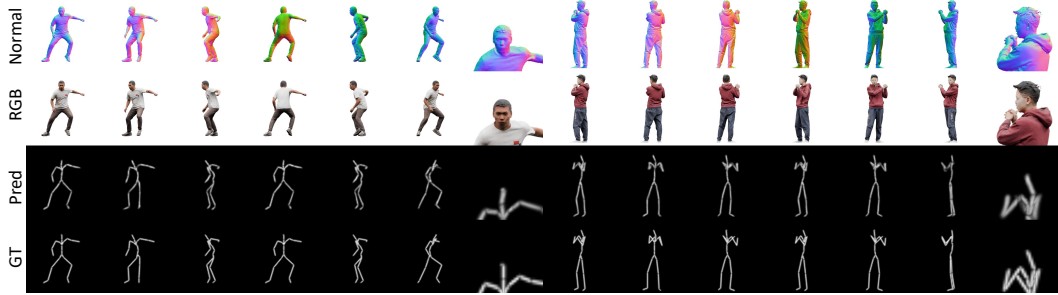

Figure 10: Visualization of $g_{skel}$ in PoseScore. $g_{skel}$ converts the multi-view latent images encoded from the normal and RGB images using the base model's VAE.

**Limitations** Similar to previous single-image-to-3D human modeling approaches, our pipeline requires segmented input images. When input images contain imperfect segmentation, artifacts such as floating geometry appear in the boundary regions of the generated 3D humans, as illustrated in Figure 11.

Although DrPose employs gradient stopping and gradient checkpointing techniques, it requires substantial GPU memory, as it generates 24 images of size 768×768 px through an iterative denoising process to compute PoseScore. Also, our KL-divergence term incurs additional GPU memory and computation overhead, as it requires storing the initial denoising U-Net and performing noise prediction at each iteration, although it mitigates reward hacking. We expect that efficiency improvements in future multi-view diffusion models will alleviate these issues.

ACKNOWLEDGEMENTS

This work was supported by IITP grant (No. RS-2021-II211343, Artificial Intelligence Graduate School Program at Seoul National University) (5%), NRF grant (No.2023R1A1C200781211) (50%), and IITP grant (No.RS-2025-02303703, Realworld multi-space fusion and 6DoF free-viewpoint immersive visualization for extended reality) (45%), funded by the Korea Government (MSIT).

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

# A APPENDIX

## A.1 ADDITIONAL RESULTS

We provide additional qualitative comparisons in Figures 15 and 16.

## A.2 LIMITATIONS

Our pipeline inherits limitations from prior single-image-to-3D approaches, shown in Figure 11. First, imperfect input segmentation causes floating geometry artifacts at boundaries. Second, while improving overall shape and pose, our method struggles with fine details like hands.

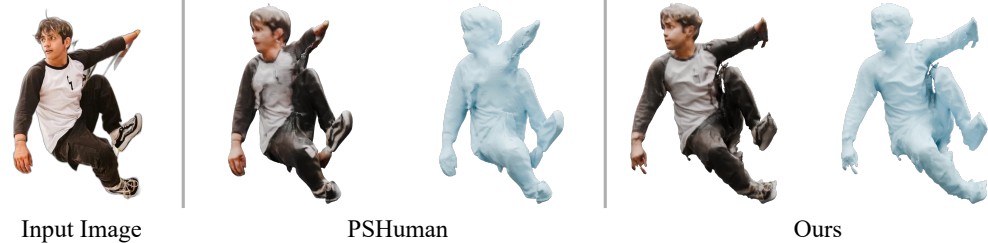

Input Image              PSHuman                          Ours

Figure 11: Our pipeline is sensitive to the quality of the segmented masks, producing artifacts.

## A.3 ANALYSIS ON EFFICIENCY

We report the latency of each model for reconstructing a single sample in Table 4. Ours has the same efficiency as PSHuman since we use it as our base model, while Ours (Era3D) has the same efficiency as the Era3D.

Table 4: Reconstruction latency per sample for each model.

| ECON | SiTH | Era3D | PSHuman | Ours (Era3D) | Ours |
|---|---|---|---|---|---|
| 183.27 sec. | 117.35 sec. | 15.24 sec. | 42.70 sec. | 15.24 sec. | 42.70 sec. |

## A.4 NETWORK ARCHITECTURE

Figures 12, 13, and 14 show the network design for the Denoising U-Net in the multi-view diffusion used in the pipeline illustrated in Fig 5, the Denoising U-Net in the MIMO Men et al. (2025), a pose-conditioned video generator, and the skeletal image predictor in the PoseScore introduced in Sec 3.1.

## A.5 USE OF LARGE LANGUAGE MODELS

We employ a large language model to refine the manuscript text by correcting grammatical errors and enhancing sentence fluency. The LLM is not involved in research ideation, methodology development, experimental design, or the generation of original content. All intellectual contributions, including the research direction, analyses, and conclusions, are made entirely by the authors.

## A.6 ETHICS STATEMENT

**Demographic Bias** Our base model, PSHuman (Li et al., 2024b), is trained on THuman2.1 (Yu et al., 2021) and CustomHumans (Ho et al., 2023b), which exhibit demographic imbalances. THuman2.1 contains 2,445 human subjects who are predominantly of Asian ethnicity, while CustomHumans, though more ethnically diverse, comprises only 647 subjects. This imbalanced representation may result in biased reconstruction performance that favors demographics overrepresented in the training data, leading to reduced quality and accuracy for underrepresented groups.

**Potential for Misuse**     The generated 3D human models pose risks for creating misleading or harmful content. These reconstructions can be integrated into 3D scenes and animated using standard rigging techniques, potentially enabling the creation of disinformation or deepfake content.

**Industrial Impact**     The automation capabilities of image-to-3D human modeling technology may impact employment in creative industries, affecting 3D artists, character designers, and digital content creators who specialize in human modeling. While this technology can enhance productivity and accessibility, it also raises questions about the displacement of skilled professionals.

## A.7    REPRODUCIBILITY STATEMENT

DrPose15K is constructed from the publicly available Motion-X dataset (Lin et al., 2023) and MIMO model (Men et al., 2025). MixamoRP is constructed from scans of RenderPeople (Renderpeople, 2023) and motions from Mixamo (Inc., 2025); while both resources are available, RenderPeople is a commercial product. In Section 3, we explain the high-level concepts underlying our approach and provide pseudocode and experimental details in Algorithm 1 and Section 4.1 to ensure reproducibility.

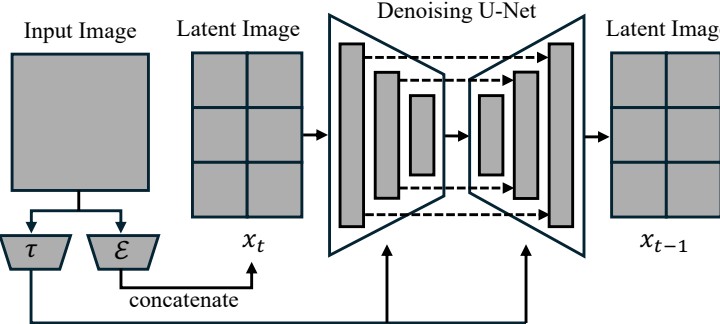

Figure 12: Architecture of the Denoising U-Net for multi-view diffusion in our pipeline inspired by Li et al. (2024b) (illustrated in Fig 5). The denoising U-Net follows the architecture of PSHuman (Li et al., 2024b). The input image is conditioned into the denoising process through two parallel pathways: (1) A VAE encoder $\mathcal{E}$ encodes the input image, which is then concatenated with the latent image $x_t$. (2) A CLIP image encoder $\tau$ encodes the input image, and the generated tokens are fed into the cross-attention layers of the denoising U-Net.

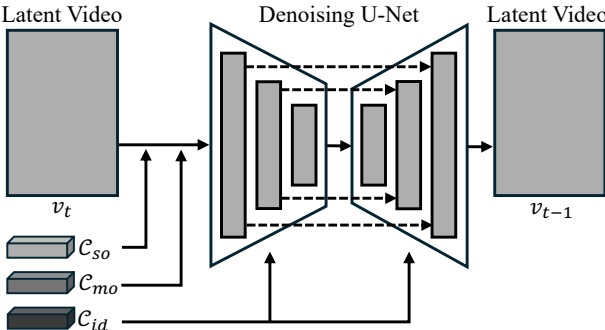

Figure 13: Architecture of the denoising U-Net for the pose-conditioned image-to-video model used in the DrPose15K construction process (illustrated in Fig. 3). The denoising U-Net follows the architecture of MIMO (Men et al., 2025) and takes three conditioning signals: (1) scene code $\mathcal{C} * \text{so}$, (2) motion code $\mathcal{C} * \text{mo}$, and (3) identity code $\mathcal{C} * \text{id}$. The scene code $\mathcal{C} * \text{so}$ is first concatenated with the latent video $v_t$ and then added to the motion code $\mathcal{C} * \text{mo}$. The identity code $\mathcal{C} * \text{id}$ is fed into the cross-attention layers of the denoising U-Net. Note that the temporal layers of the denoising U-Net (Guo et al., 2023) are omitted in this figure.

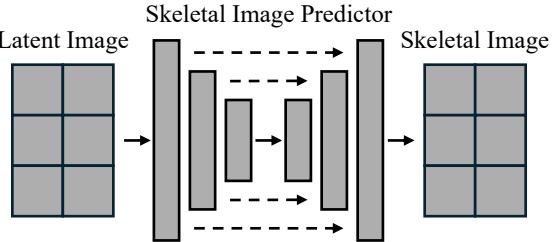

Figure 14: Architecture of the Skeletal Image Predictor in our PoseScore introduced in Sec 3.1. The network follows a U-Net architecture with an encoder-decoder structure. Multi-view latents are processed through initial convolutions, then flattened and passed through four downsampling blocks (reducing spatial resolution from 64×64 to 4×4 while increasing channels from 32 to 512), followed by four upsampling blocks with skip connections that restore the original resolution. The output produces predicted skeletal images for all views simultaneously.

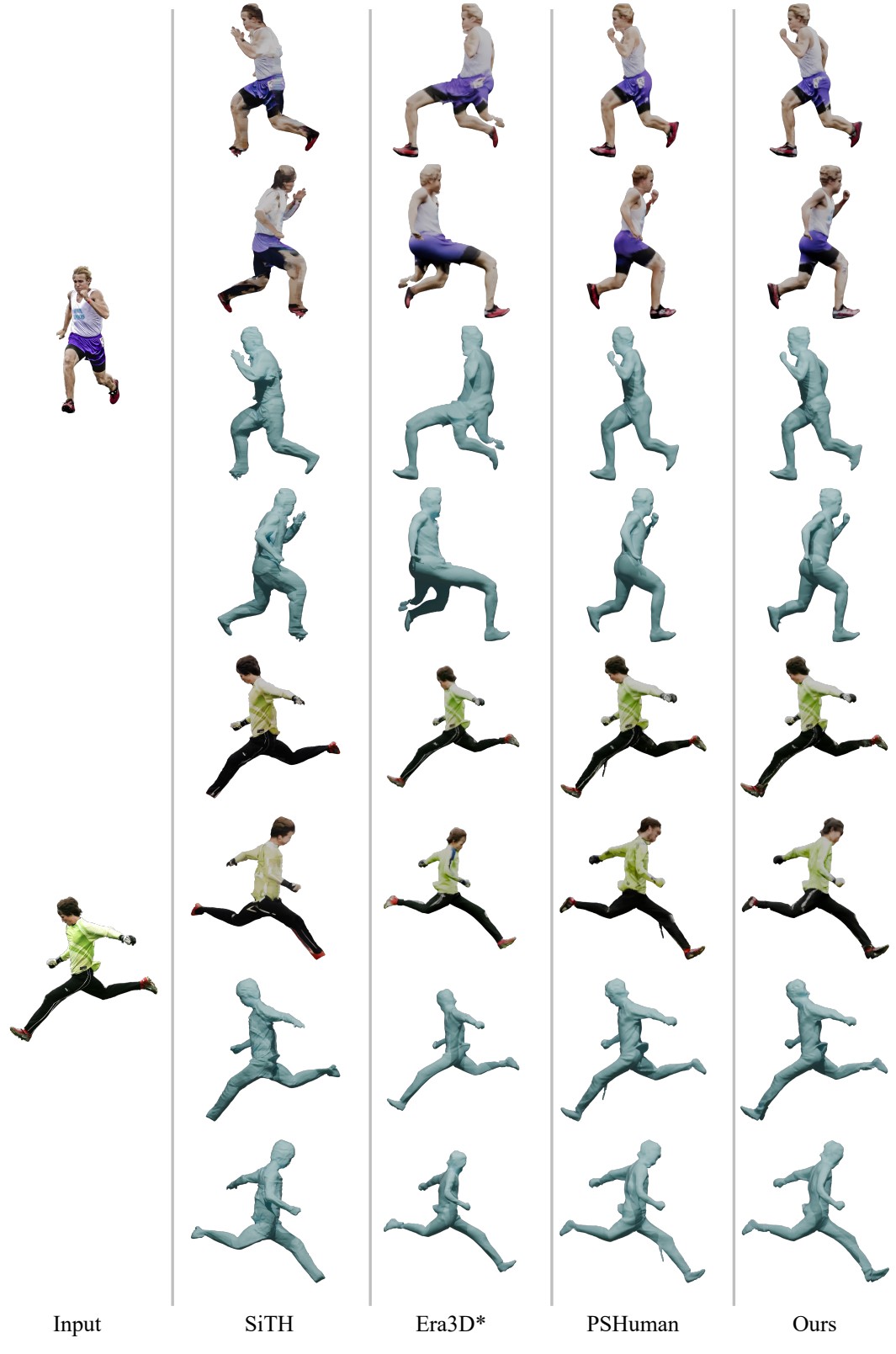

Figure 15: Additional qualitative evaluation on the internet-source images. Era3D* denotes Era3D fine-tuned on CustomHumans and THuman2.1 datasets.

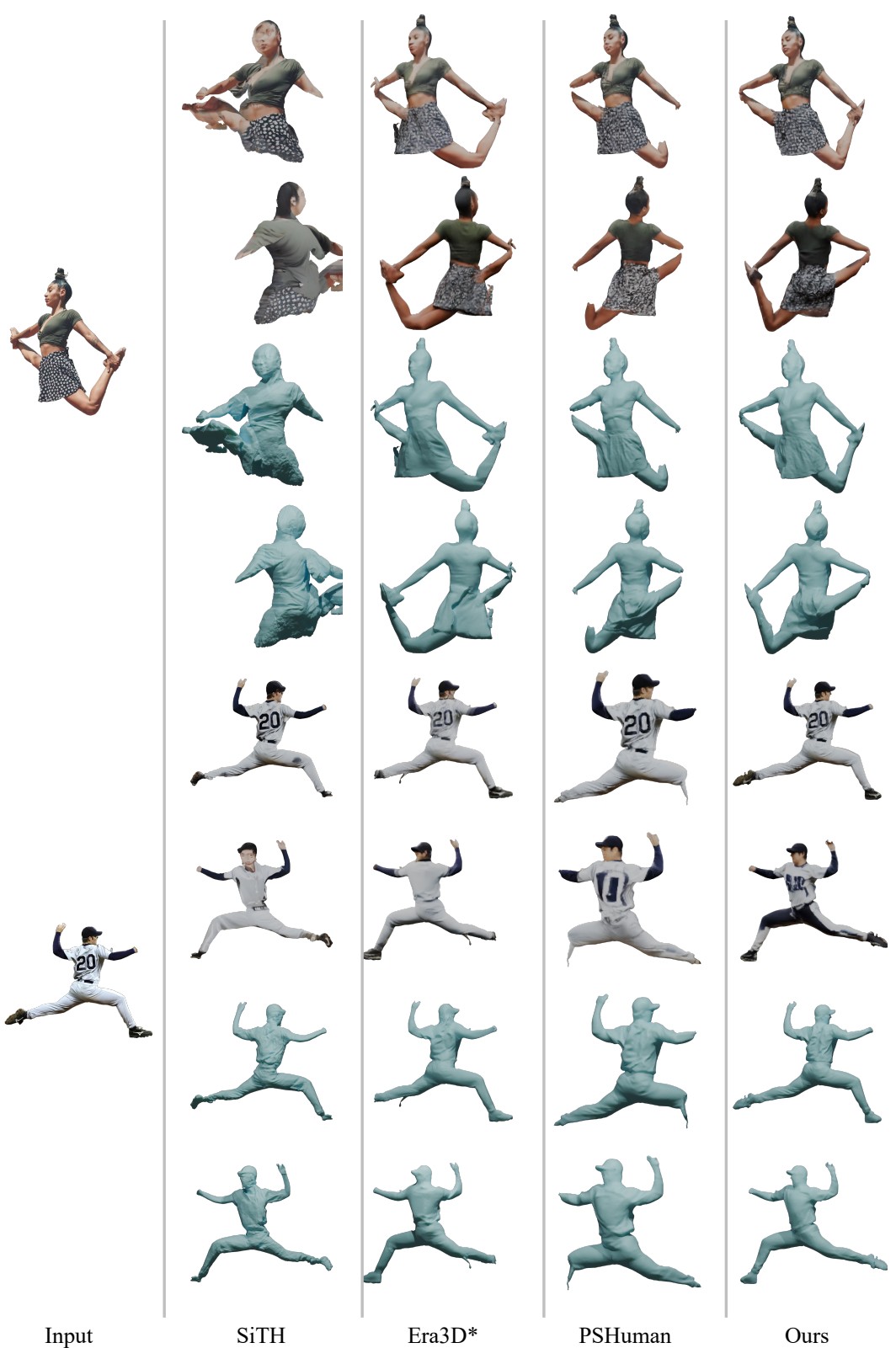

Input          SiTH          Era3D*          PSHuman          Ours

Figure 16: Additional qualitative evaluation on the internet-source images. Era3D* denotes Era3D fine-tuned on CustomHumans and THuman2.1 datasets.

Table 5: MixamoRP dataset specification

| Character | Animation | Description | Frame number(s) |
|-----------|-----------|-------------|-----------------|
| Carla | Drop Kick | - | 35, 46, 62 |
| Carla | Start Plank | - | 137 |
| Claudia | Freehang Climb | - | 47, 67 |
| Claudia | Flying Knee Punch Combo | - | 29, 79 |
| Eric | Swing To Land | Swing Backflip To Crouched Land | 26, 58 |
| Eric | Standing Up | Sitting To Standing | 41, 88 |
| Henry | Situp To Idle | - | 15, 49, 70 |
| Henry | Female Standing Pose | On Left Leg, Right Hand... | 1 |
| Johanna | Twist Dance | - | 163 |
| Johanna | Jump Push Up | - | 25 |
| Johanna | Sitting Laughing | - | 67 |
| Johanna | Praying | ...Prayer To Standing Up | 1 |
| Kumar | Rifle Turn And Kick | - | 40, 48 |
| Kumar | Dancing Twerk | - | 179 |
| Kumar | Crouch Turn Left 90 | Turning 90 Degrees Left | 6 |
| Michael | Pain Gesture | - | 20 |
| Michel | Breakdance 1990 | ...Handstand Spin Start | 1, 82, 100 |
| Mira | Change Direction | - | 25 |
| Mira | Mma Kick | Mma Medium Kick | 15, 22 |
| Mira | Beckoning | - | 26 |
| Otto | Throw Grenade | ...While In Prone Position | 65 |
| Otto | Run Backwards | ...Backwards To Crouched Stop | 37 |
| Otto | Hurricane Kick | - | 16 |
| Otto | Grabbing Ammo | - | 74 |
| Sebastian | Pistol Kneeling Idle | - | 1 |
| Sebastian | Crawling | - | 34 |
| Sebastian | Dig And Plant Seeds | - | 15, 70 |
| Sheila | Shuffling | - | 33 |
| Sheila | Great Sword Slash | Great Sword Combo Slash | 47, 55, 62 |
| Sydney | Sword And Shield Attack | Sword And Shield High Attack | 17, 26 |
| Sydney | Running Jump | Jumping From A Sprint | 7, 22 |
| Tiffany | Samba Dancing | Afoxe Samba Reggae Dance | 139 |
| Tiffany | Stable Sword Inward Slash | - | 5, 27 |
| Toshiro | Martelo 2 | - | 17 |
| Toshiro | Jump Attack | - | 11, 27, 53 |
| Victoria | Great Sword Crouching | ...Sword Crouch To Block | 10 |
| Victoria | Chapa-Giratoria | - | 61 |
| Victoria | Jab Cross | Boxing Jab Cross Medium | 22 |
| Victoria | Jump | Jump In Place | 35 |

