# OpenReview forum: "Direct Reward Fine-Tuning on Poses for Single Image to 3D Human in the Wild"
_ICLR.cc/2026/Conference — ICLR 2026 Poster_

### Official Review · Reviewer_uquA · 2025-10-30

**Soundness:** 2
**Presentation:** 2
**Contribution:** 2
**Rating:** 4
**Confidence:** 5

**Summary:**

This paper addresses the challenge of reconstructing 3D humans from single images, specifically focusing on "hard poses" (e.g., dynamic or acrobatic) where existing methods often fail. The paper's main contributions are twofold: 1) The proposal of DRPOSE15K, a new synthetic dataset constructed using an existing motion dataset and a pose-conditioned video generative model, which is designed to have higher pose diversity than previous datasets. 2) The introduction of DRPOSE, a direct reward fine-tuning algorithm that post-trains a multi-view diffusion model to better align with the diverse poses in the new dataset. The stated goal is to improve the quality and pose accuracy of 3D human reconstructions, particularly for these challenging in-the-wild cases.

**Strengths:**

- **Task:** The paper focuses on "hard poses," which is a core, unsolved problem for most single-image 3D human reconstruction methods.
- **Dataset:** The creation of the DRPOSE15K dataset, which features a measurably higher diversity of poses compared to existing 3D human datasets, is a valuable and positive contribution to the field.
- **Idea:** The core concept of using direct reward fine-tuning to improve a multi-view generator's alignment with specific poses is intuitive.

**Weaknesses:**

**Major:**

1. **Questionable Novelty & Base Model Choice:**
The paper's core motivation is undermined by the choice of MIMO as the base pose-conditioned video generator, which produces results with poor multi-view consistency. This is surprising, as other recent models cited in the paper (e.g., the finetuned-MVChamp in IDOL (Instant Photorealistic 3D Human Creation; CVPR’25), original Champ, MimicMotion, SV3D) appear to offer superior texture and consistency. This raises a critical question: **If a stronger, state-of-the-art video generator were used, would the proposed DRPOSE fine-tuning still be necessary?** To validate the generality and necessity of the reward fine-tuning, the authors should test their method on a stronger video generation backbone (e.g., Champ) and present additional results.

2. **Missing Critical Baseline:** As mentioned before, the paper fails to compare against IDOL, which is a highly relevant and important baseline. IDOL also employs a similar concept of data synthesis using a generative video model and achieves state-of-the-art results.
3. **Extremely Poor Qualitative Results:** The visual quality in the teaser (Fig. 1) and other figures shows severe texture and color inconsistency across views.
4. **Missing Ablation Study:**
    - The primary technical contribution is the reward fine-tuning (DRPOSE) to improve the multi-view generator. However, the paper lacks an ablation study that *isolates* the effect of this fine-tuning. There should be a direct evaluation (using video generation metrics and pose alignment metrics) of the multi-view generator *before* and *after* the DRPOSE fine-tuning, disentangled from the final reconstruction results, to prove its efficacy.
    - The paper does not provide an ablation study (either qualitative or quantitative) on the impact of the KL divergence regularization term ($L_{KL}$). I’m wondering how much this component contributes or what happens if it is removed.
5. **Dataset Realism:** The textures on the synthesized humans in DRPOSE15K look non-realistic. Does this lack of texture fidelity in the training data limit the model's ability to generalize and produce realistic textures for in-the-wild test images?

**Minor:**

1. **Inadequate Literature Review:** The paper fails to cite or discuss an important category of recent, high-quality optimization-based methods (SDS-based), including:
    - TeCH: Text-guided Reconstruction of Lifelike Clothed Humans
    - Human-SGD: Single-Image 3D Human Digitization with Shape-Guided Diffusion
    - GeneMAN: Generalizable Single-Image 3D Human Reconstruction from Multi-Source Human Data
    - Relevant Gaussian-based reconstruction work, such as the aforementioned IDOL, is also omitted.
2. **Methodological Unclarity & Typos:**
    - The method description is unclear at points (e.g., the sudden introduction of $x_0$ in L295 without clear definition).
    - L174: "kmage" should be "image-to-multi-view"
    - L231: "Algrotihm" should be "Algorithm"
3. **Insufficient Efficiency Analysis:** In Appendix A.4, a total latency is reported. This would be more useful as a detailed time breakdown for the two main stages: 1) multi-view video generation and 2) human carving/reconstruction.

**Questions:**

1. Why was a 2D skeleton-image-based alignment chosen for the pose reward? Why not use a more direct 3D-based reward, such as using a pre-trained HMR model to estimate SMPL parameters from the generated images and then applying a 3D joint loss? Is there an ablation study demonstrating that the skeleton-based approach is superior?
2. Why do the quantitative and qualitative results for the PSHuman baseline appear significantly worse in this paper than those presented in the original PSHuman paper and its official project page?

---

> ### Author Response · Authors · 2025-11-24
>
> Thank you for recognizing our work's motivation and contributions to datasets, and the intuitiveness of DrPose, as well as pointing out the weaknesses and raising several essential questions.
>
> Below are our responses to the reviewer's concerns and questions:
>
> ***1. Questionable Novelty & Base Model Choice***
>
> Before beginning the main discussion, we hope to clarify the following:
> - The texture qualities of Champ and Mimic-motion are not better than MIMO, as shown in Fig6 and Table1 of MIMO[1].
> - Existing pose-conditioned video generators, including both MIMO and Champ are hard to be directly used to single-view 3D human reconstruction since they are not multi-view consistent enough. In Sec3.2 of IDOL[2] the authors mentioned that:
>
> > Image-based animating models struggle to achieve the generation of 360 consistent and diverse human videos conditioned on pose sequences (e.g., 2D poses and 3D SMPL(-X) conditions)
>
> ***If a stronger, state-of-the-art video generator were used, would the proposed DRPOSE fine-tuning still be necessary?***
>
> While adopting a video generator instead of a multi-view diffusion model is a promising approach, we argue that existing pose-conditioned video generators at the time of submission suffer from multi-view inconsistency issues that limit their direct application to single-view 3D human reconstruction. This limitation persists even in MV-Champ, a pose-conditioned video generator specifically fine-tuned on human scan datasets to improve multi-view consistency.
>
> The multi-view inconsistencies are observable in the generated images through a simple validation. For instance, examining the height of humans across different viewpoints reveals discrepancies. In the “Comparison with MV-Champ on HuGe100K" section of the [website](https://somerandomuser1117.github.io/drpose_rebuttal.github.io/), we demonstrate these limitations.
>
> For comparison, we also show the multi-view images generated using our model. We select a single input image among the 24 multi-view images of a sample in the HuGe100K, then we generate multi-view images using our model. In these generated images, the human has the same height across all views.
>
> Returning to the original question regarding whether stronger state-of-the-art video generators would eliminate the need for our proposed DrPose fine-tuning, we argue that DrPose remains valuable because existing pose-conditioned video generators lack sufficient multi-view consistency for reliable 3D reconstruction applications.
> The reviewer also suggested training a pose-conditioned video generator using DrPose directly. While this is a promising research direction, we expect that such an approach would not surpass our current pipeline. This is because DrPose enhances pose accuracy but does not address the multi-view consistency limitation. Ideally, we would validate this hypothesis through an experiment; however, the time constraints of the rebuttal period preclude such extensive experimentation.
>
> ***2. Comparison with IDOL***
>
> We compare our method with IDOL below and in the section “Comparison with MV-Champ” of the [website](https://somerandomuser1117.github.io/drpose_rebuttal.github.io/):
>
> Quantitative evaluation on CustomHumans
> |              | PSNR &uarr;  | SSIM &uarr;  | LPIPS &darr; |
> | ------------ | ------- | ------ | ------ |
> | IDOL | 16.7608 | 0.8110 | 0.1935 |
> | Ours | 19.3404 | 0.8411 | 0.1224 |
>
> |              | PSNR &uarr;  | SSIM &uarr;  | LPIPS &darr;  |
> | ------------ | ------- | ------ | ------ |
> | IDOL    | 16.5866 | 0.8121 | 0.1690 |
> | Ours | 19.3110 | 0.8303 | 0.1243 |
>
> Note:
> - Although there is a large margin in both quantitative comparison in the table above and the qualitative comparison in the section “Comparison with IDOL” of the [website](https://somerandomuser1117.github.io/drpose_rebuttal.github.io/), this gap is mainly attributable to the backbone, as PSHuman provides a stronger baseline than IDOL.
> - Since IDOL does not provide the evaluation code, we designed an evaluation protocol in our submission. We normalize the results into a unit cylinder for fair comparison since each method has different scales. However, since the IDOL sometimes generates unbounded 3DGS with floaters, this normalization makes the result’s scale smaller. For this reason, we employ OpenPose to detect 2D keypoints, compute the affine transformation from results to input images, and warp them to obtain normalized samples.
> - Since the output format of IDOL is 3DGS, we primarily focus on pixel-level reconstruction metrics, as it’s not straightforward to compute geometric metrics(Chamfer Distance, Normal Consistency, and F-score).

---

> ### Author Response · Authors · 2025-11-24
>
> ***3. Rendering Issue in the Qualitative Results***
>
> As mentioned in Section 2 of the global response, the rendering parameter choices caused the colors to appear washed out. We have corrected this issue, as displayed in the section “Correction on rendering results” of the [website](https://somerandomuser1117.github.io/drpose_rebuttal.github.io/).
>
> ***4. Dataset Realism***
>
> The textures of samples in DrPose15K are unrealistic (though they are plausible), as the dataset primarily consists of extreme and diverse poses. However, as discussed in Section 1 of the global response, this is not critical to our algorithm since we optimize the multi-view diffusion model using PoseScore, which computes the consistency between the generated latents and ground-truth poses. In contrast, previous methods that train multi-view diffusion in an unsupervised manner using DDPM loss can be critically affected by texture quality, which relates to our discussion in Section 2 of this response.
>
> ***5. Suggestions on the paper and Report of typos***
>
> Thank you for your suggestions. We have incorporated the following changes in the revised version:
> Added references (TeCH, GeneMAN, Human-SGD, IDOL)
>
> - Fixed typo: "kmage" should be "image-to-multi-view"
> - Fixed typo: "Algrotihm" should be "Algorithm"
>
> ***6. Why do the quantitative and qualitative results for the PSHuman baseline appear significantly worse in this paper than those presented in the original PSHuman paper and its official project page?***
>
> This is due to the difference in the details in the evaluation protocol, including normalization and which views are used for the input images for each sample in the test dataset. As we couldn’t find the exact evaluation protocol for PSHuman (the evaluation code is not provided, and the specific protocol procedure is not specified in the paper), we used the following protocol. For normalization, we used a commonly used technique for point clouds with different scales: translating to their centroid and scaling them to fit within a unit sphere or unit cylinder. For input image generation, we used three evenly distributed viewpoints for each test sample to ensure model evaluation across diverse views.
>
> [1] Men, Yifang, et al. "Mimo: Controllable character video synthesis with spatial decomposed modeling." Proceedings of the Computer Vision and Pattern Recognition Conference. 2025.
>
> [2] Zhuang, Yiyu, et al. "Idol: Instant photorealistic 3d human creation from a single image." Proceedings of the Computer Vision and Pattern Recognition Conference. 2025.

---

### Official Review · Reviewer_eWSd · 2025-10-31

**Soundness:** 3
**Presentation:** 3
**Contribution:** 3
**Rating:** 6
**Confidence:** 3

**Summary:**

This paper presents DRPOSE, a framework designed to improve single-image 3D human reconstruction, particularly for challenging poses. The authors identify limitations in existing datasets and corresponding methods for training and evaluating models for this task. To address these issues, they introduce a post-training approach for image-to-multi-view (I2MV) diffusion models using a differentiable reward function, POSESCORE, to better align the model outputs with ground-truth poses. Furthermore, they contribute DRPOSE15K, a diverse pose dataset for training, and MIXAMORP, a benchmark for evaluating performance on extreme poses. The proposed system is evaluated on several datasets, showing consistent improvements in pose accuracy and reconstruction quality.

**Strengths:**

- Comprehensive Motivation and Contribution:

The paper starts with a clear motivation, highlighting the limitations of existing datasets and evaluation protocols for single-image 3D human reconstruction. To address these, the authors propose a systematic set of contributions: a novel post-training method (POSESCORE-driven fine-tuning), a new training dataset (DRPOSE15K), and a tailored evaluation benchmark (MIXAMORP). The contributions are well-aligned with the stated motivation and provide a holistic solution to the identified challenges.

- Dataset Contributions (Training and Evaluation):

DRPOSE15K significantly expands the diversity of human poses compared to prior datasets like THuman2.1, offering a valuable resource for training models on challenging poses. Additionally, the MIXAMORP benchmark addresses a critical gap in evaluating reconstruction performance under extreme pose variations, making it a practical addition for future research in this area.

**Weaknesses:**

- Pipeline Novelty:

While the paper provides a systematic solution, the overall pipeline lacks substantial innovation. The framework is primarily built on an image-to-multi-view (I2MV) diffusion model followed by existing 3D reconstruction techniques. The pipeline primarily follows previous designs(Like PSHuman), with the main novelty lying in the reward-based fine-tuning (POSESCORE) and the dataset contributions.

- Limited Impact of Individual Contributions:

The reward function POSESCORE, while effective, is relatively straightforward and lacks deeper theoretical exploration. Similarly, the dataset and benchmark contributions, while useful, are incremental rather than transformative.

Overall, this is a well-executed paper, though not a groundbreaking one.

**Questions:**

Your method defines the reward as r(x_0, θ) but uses it to derive the loss function L_reward = 1 - r(x_0, θ). Why do you choose to frame it as a reward instead of directly treating it as a loss?

---

> ### Author Response · Authors · 2025-11-24
>
> Thank you for recognizing the motivation and contributions of our work regarding a novel post-training method, training dataset, and benchmark, as well as for pointing out weaknesses of our work.
>
> Here we respond to the reviewer's concerns:
>
> ***1. Pipeline novelty & Limited Impact of Individual Contributions***
>
>  Our framework adopts direct-reward fine-tuning to improve the reconstructed poses in human multi-view diffusion models, which hasn’t been explored in the human reconstruction field previously.
>
> While existing approaches inspire some components, our main contribution lies in integrating these components to enhance single-view 3D human reconstruction. To our knowledge, DrPose is the first attempt to apply direct reward fine-tuning methods (originated from reinforcement learning) to multi-view diffusion for enhancing human poses. While there have been efforts to improve aesthetic quality or text-alignment in other domains, such as image generation, this represents a novel application to human pose enhancement. Although PoseScore may sound conceptually straightforward, we believe it is a well-designed differentiable reward model for measuring consistency between generated multi-view images and ground truth poses, as evidenced by its strong performance across three benchmarks.
>
> ***2. Your method defines the reward as r(x_0, θ) but uses it to derive the loss function L_reward = 1 - r(x_0, θ). Why do you choose to frame it as a reward instead of directly treating it as a loss?***
>
> As reviewer iYqC mentioned, our DrPose approach has its origins in reinforcement learning (RL). While DrPose cannot be strictly classified as RL, given that our PoseScore is a differentiable function, the direct reward fine-tuning methods it follow (DrTune[1], AlignProp[2]) are derived from RL frameworks. Therefore, we chose to follow RL conventions.
>
> [1] Wu, Xiaoshi, et al. "Deep reward supervisions for tuning text-to-image diffusion models." European Conference on Computer Vision. Cham: Springer Nature Switzerland, 2024.
>
> [2] Prabhudesai, Mihir, et al. "Aligning text-to-image diffusion models with reward backpropagation." (2023).

---

### Official Review · Reviewer_hy1Y · 2025-11-01

**Soundness:** 3
**Presentation:** 3
**Contribution:** 2
**Rating:** 4
**Confidence:** 5

**Summary:**

This paper tackles the problem of unnatural poses in single-view 3D human reconstruction, particularly for dynamic and challenging movements. The authors propose DRPOSE, a direct reward fine-tuning method for image-to-multi-view (I2MV) diffusion models that doesn't require expensive 3D human assets. Key contributions include:

DRPOSE15K: A dataset of 15K diverse poses paired with single-view images, constructed from Motion-X motion data and the MIMO pose-conditioned video generator

DRPOSE Algorithm: Post-training method using a differentiable POSESCORE reward that measures consistency between generated multi-view latents and ground-truth poses

MIXAMORP Benchmark: New evaluation benchmark for challenging human poses
Consistent improvements across benchmarks (THuman2.1, CustomHumans, MIXAMORP)

**Strengths:**

1.Well-motivated focus on pose quality limitations in existing approaches, with quantitative evidence (1.73× larger pose diversity in DRPOSE15K vs THuman2.1)

2.Creative data construction: Leveraging motion capture + video generation to avoid expensive 3D scanning.

**Weaknesses:**

Circular dependency in data quality: DRPOSE15K relies on MIMO to generate training images, creating a potential bottleneck. If MIMO produces unrealistic appearances for extreme poses, the model learns from flawed data. This isn't adequately discussed or validated.

Insufficient reward model analysis: no ablation on reward formulation alternatives (e.g., direct 3D keypoint prediction, discriminator-based rewards).

Insufficient failure cases are shown, it is hard to evaluate, but the geometry shown in supp videos are pool such as the artifacts in hands, legs and clothes.

Missing references:
SIFU: Side-view Conditioned Implicit Function for Real-world Usable Clothed Human Reconstruction. In CVPR 2024
TeCH: Text-guided Reconstruction of Lifelike Clothed Humans. In 3DV 2024
Generalizable Human Gaussians from Single-View Image. In ICLR 2025
Humangif: Single-view human diffusion with generative prior. In Arxiv 2025
WonderHuman: Hallucinating Unseen Parts in Dynamic 3D Human Reconstruction. In Arxiv 2025

**Questions:**

Would some samples of DRPOSE15K be shown in to see the quality? Would this be open-sourced?
How different is the proposed method compared to MagicMan which also finetunes MV-diffusion on human dataset?
How loose clothes is handled, how appearance is handled?

---

> ### Author Response · Authors · 2025-11-24
>
> We thank the reviewer for their sincere and helpful comments. We appreciate the reviewer's recognition that our work is well-motivated and that the suggested data construction approach is creative.
> Below are our responses to the reviewer's concerns and questions:
>
> ***1. On the Unrealistic Appearance of MIMO-Generated Images for Extreme Poses***
>
> While textures of DrPose15K are usually plausible, we agree that they can be unrealistic, as the dataset primarily consists of extreme and diverse poses. However, as discussed in Section 1 of the global response, this is not critical to our algorithm since we optimize the multi-view diffusion model using PoseScore, which computes the consistency between the generated latents and ground-truth poses. In contrast, previous methods that train multi-view diffusion in an unsupervised manner using DDPM loss can be affected by texture quality, which relates to our discussion in Section 2 of this response.
>
> ***2. How different is the proposed method compared to MagicMan which also finetunes MV-diffusion on human dataset?***
>
> Several existing single-view 3D human reconstruction methods (MagicMan[4], PSHuman[5], Humansplat[6]; please see citations in Section 2 of the main paper) fine-tune MV-diffusion on human datasets as mentioned in the question. The key difference between our proposed DrPose and these methods is that *DrPose trains an MV-diffusion model using a differentiable reward model rather than using DDPM loss*. These existing methods require ground-truth multi-view images, which are challenging to diversify in terms of poses due to construction costs. In contrast, our algorithm requires only single-view images paired with SMPL models, and the loss is computed using a differentiable reward that measures consistency between the SMPL model and generated multi-view images. This advantage enables us to utilize existing SMPL models and pose-conditioned diffusion models that generate corresponding single-view images to construct DrPose15K, which contains diverse and extreme poses. Furthermore, the previous approaches are sensitive to texture quality in the dataset, while our approach is more robust, as discussed in Section 1 of this response.
>
> ***3. Would some samples of DRPOSE15K be shown to see the quality? Would this be open-sourced?***
>
> We have uploaded additional samples of DrPose15K in the section “Additional examples for DrPose15K” of the [website](https://somerandomuser1117.github.io/drpose_rebuttal.github.io/). We will open-source this dataset after reviewing the licenses of the dependencies (MIMO[1], Motion-X[2], and images from Generated Photos[3]).
>
> ***4. Missing references***
>
> Thank you for identifying several important missing references in the single-view 3D human reconstruction field, including a SMPL-prior-based method (SIFU), SDS-based methods (TeCH, WonderHuman), and a multi-view diffusion-based method (Humangif). We have cited these methods in Section 2 of our revised version.
>
> [1] Men, Yifang, et al. "Mimo: Controllable character video synthesis with spatial decomposed modeling." Proceedings of the Computer Vision and Pattern Recognition Conference. 2025.
>
> [2] Lin, Jing, et al. "Motion-x: A large-scale 3d expressive whole-body human motion dataset." Advances in Neural Information Processing Systems 36 (2023): 25268-25280.
>
> [3] Unique, worry-free model photos. Generated Photos. (n.d.). https://generated.photos/
>
> [4] He, Xu, et al. "Magicman: Generative novel view synthesis of humans with 3d-aware diffusion and iterative refinement." Proceedings of the AAAI Conference on Artificial Intelligence. Vol. 39. No. 3. 2025.
>
> [5] Li, Peng, et al. "PSHuman: Photorealistic Single-image 3D Human Reconstruction using Cross-Scale Multiview Diffusion and Explicit Remeshing." arXiv preprint arXiv:2409.10141 (2024).
>
> [6] Pan, Panwang, et al. "Humansplat: Generalizable single-image human gaussian splatting with structure priors." Advances in Neural Information Processing Systems 37 (2024): 74383-74410.

---

### Official Review · Reviewer_iYqC · 2025-11-01

**Soundness:** 4
**Presentation:** 4
**Contribution:** 4
**Rating:** 10
**Confidence:** 4

**Summary:**

This paper proposes a strategy to enable robust human geometry reconstruction under extreme poses. To achieve this, the DRPose15k dataset and a differentiable reward model fine-tuning strategy are introduced.

**Strengths:**

- Very well motivated — recent diffusion-based methods fail on complicated poses (e.g., acrobatic poses).

- Well-designed and effective RL-based reward model that is differentiable.

- Strong improvement over SOTA methods (SiTH, PSHuman, etc.).

- Thoroughly studied recent literature.

- The DRPose15k dataset provides extreme-pose ground-truth SMPL-image pairs, which are valuable to the community. The idea of using a pose-conditioned video diffusion model to create this dataset is brilliant, as going in the opposite direction does not work well (i.e., curating extreme-pose RGB images and then using off-the-shelf SMPL estimators).

- Evaluated on multiple datasets, validating the generalizability of this method.

- Overall, I enjoyed reading this paper. Solid results with strong motivation and a well-designed, effective RL-based strategy. I vote for the acceptance of the paper. As this paper can enlighten future works, I encourage the authors to improve the reproducibility in the final revision by providing more implementation details.

**Weaknesses:**

- Colors seem washed out; I have noticed this in recent diffusion-based single-view human reconstruction methods. Why is this happening, and how can it be improved?

- Reproducibility concern: As suggested in the “Suggestions” section, it would be great if architectural details could be included in the appendix of the final revision.

- Otherwise, I am happy with the current submission.

**Questions:**

- Which pose-conditioned image-to-video diffusion model was used to create the DRPose15k dataset?

- What happens if the pose-conditioned image-to-video diffusion model fails for extreme poses? Do you manually filter out those noisy outputs? How do such outliers affect model performance?

- Typo in Fig. 2 caption: “Kmage-to-multi-view” → “Image-to-multi-view.”

- Suggestion: In Fig. 5, add a brief explanation of the algorithm. The figure should be self-explanatory.

- Suggestion: Please include architectural details (e.g., video diffusion model, denoising UNet, skeleton decoder, etc.) in the appendix for reproducibility.

---

> ### Author Response · Authors · 2025-11-24
>
> We thank the reviewer for the sincere and helpful comments. We appreciate the reviewer's recognition that our work is well-motivated and that the suggested RL-based differentiable reward model is well-designed.
> Here are our responses to the reviewer's concerns and questions:
>
> ***1. Rendering Issue in the Qualitative Results***
>
> As mentioned in Section 2 of the global response, the rendering parameter choices caused the colors to appear washed out. We have since corrected this issue, as displayed in the section “Correction on rendering results” of the [website](https://somerandomuser1117.github.io/drpose_rebuttal.github.io/).
>
> ***2. What happens if the pose-conditioned image-to-video diffusion model fails for extreme poses?***
>
> The possible failure scenario can happen in two cases: (1) unrealistic textures and (2) pose misalignment with paired SMPL models. The first case is not critical to our algorithm since we optimize the multi-view generation using PoseScore, which computes the consistency between the generated latents and ground-truth poses, as discussed in Section 1 of the global response. The second type of failure could be critical to our model; however, we did not observe such shortcomings in our DrPose15K dataset when we randomly sampled and examined 2,000 samples from the complete dataset. If some poses are inconsistent with the images, we expect model accuracy to decrease due to incorrect gradient flow from mismatched data pairs.
>
> ***3. Do you manually filter out those noisy outputs?***
>
> For the generated images, we didn’t conduct manual filtering. But we conducted manual filtering on the SMPL models before generation as described in Section 4.1 of the main paper. Motion-X[2] consists of several subsets of motion datasets. Among them, we used only the AIST subset since we found empirically that other subsets often contain noisy SMPL models with poses that are either physically impossible for humans to perform or where limbs intersect with other body parts.
>
> ***4. Which pose-conditioned image-to-video diffusion model was used to create the DRPose15k dataset?***
>
> We used MIMO [1] as the pose-conditioned image-to-video diffusion model. We have added this citation to the caption in the revised version for better clarity.
>
> ***5. Suggestions on the paper and Report of typos***
>
> Thank you for your suggestions and report of typos in the paper. We have incorporated the following changes in the revised version:
> - A brief explanation of the DrPose in Fig. 5.
> - A Typo: Kmage-to-multi-view” → “Image-to-multi-view.”
>
> We will include a section detailing the architecture in the appendix during the rebuttal period.
>
> [1] Men, Yifang, et al. "Mimo: Controllable character video synthesis with spatial decomposed modeling." Proceedings of the Computer Vision and Pattern Recognition Conference. 2025.
>
> [2] Lin, Jing, et al. "Motion-x: A large-scale 3d expressive whole-body human motion dataset." Advances in Neural Information Processing Systems 36 (2023): 25268-25280.

---

### Author Response · Authors · 2025-11-24
**Global Response**

We thank all reviewers for their constructive feedback and for recognizing the strengths of our work. We are particularly grateful that all reviewers found our paper well-motivated and acknowledged our contribution to addressing the limitations of recent diffusion-based methods on complex poses. We also appreciate the valuable questions raised, which have helped strengthen our paper.

The common concerns regarding our submission can be summarized as follows: (1) the texture quality of DrPose15K, and (2) the washed-out textures in our qualitative results. In this global response, we address these concerns before responding to individual reviewer comments.

We have created a [website](https://somerandomuser1117.github.io/drpose_rebuttal.github.io/) for this rebuttal to provide visual materials.

***1. Concern on Texture quality of DrPose15K which can impact on the performance of trained model (hy1Y, uquA)***

Reviewers expressed concerns that if MIMO produces unrealistic appearances for extreme poses, the model might learn from flawed data, potentially limiting its ability to generalize and produce realistic textures for in-the-wild data.

Regarding this concern, we emphasize that the primary strength of DrPose15K lies in its inclusion of diverse poses with plausible textures. While the samples are synthetic, the overall quality remains plausible (see additional DrPose15K samples on our [website](https://somerandomuser1117.github.io/drpose_rebuttal.github.io/).

Furthermore, DrPose is robust to texture imperfections due to its training approach. Unlike previous methods such as IDOL[1] and MagicMan[2], which directly fine-tune multi-view diffusion models to produce images similar to ground truth images (making them susceptible to dataset texture quality), our approach trains the model using PoseScore to optimize for multi-view consistency with ground truth poses rather than visual similarity to training images. By penalizing pose-level misalignment instead of RGB-level feature differences, our strategy decouples texture quality from the training data, allowing the model to be robust against imperfect, synthetic textures. This robustness is demonstrated in Figures 6 and 10-11 (main paper and appendix), which show the model's ability to generate realistic textures even for in-the-wild inputs.

***2. Addressing the issue where the rendered textures in the qualitative results seem washed out (iYqC, uquA)***

Thanks for pointing out the rendering issue in our qualitative results. We found its cause to be the combination of rendering parameters, including exposure, roughness, and index of refraction (IOR) values used when rendering the final meshes with vertex colors.
We have recalibrated these rendering parameters and updated the visualizations on the anonymous project [website](https://somerandomuser1117.github.io/drpose_rebuttal.github.io/) for comparison. All renderings in the paper will be revised with improved parameter settings during the rebuttal period.

[1] Zhuang, Yiyu, et al. "Idol: Instant photorealistic 3d human creation from a single image." Proceedings of the Computer Vision and Pattern Recognition Conference. 2025.

[2] He, Xu, et al. "Magicman: Generative novel view synthesis of humans with 3d-aware diffusion and iterative refinement." Proceedings of the AAAI Conference on Artificial Intelligence. Vol. 39. No. 3. 2025.

---

### Author Response · Authors · 2025-12-03

Dear AC,

Thank you for your time and consideration in reviewing our work this year. We appreciate the valuable feedback from all reviewers and have carefully addressed their concerns. We sincerely request that you have a look at **our response to each reviewer** and the **updated main paper**.

Below is a summary of key resolutions by the reviewer:

**Reviewer iYqc:** We have resolved the reviewer's concerns by: (1) correcting the rendering issues in our qualitative results, and (2) adding detailed sections describing the network architectures.

**Reviewer hy1Y:** We have provided clarification regarding the texture quality concerns for DrPose15K and added additional examples to our supplementary website ([link](https://somerandomuser1117.github.io/drpose_rebuttal.github.io/)).

**Reviewer eWSd:** Regarding the novelty of individual components, we have emphasized that our main contribution lies in adopting direct-reward fine-tuning to improve reconstructed poses in human multi-view diffusion models, an approach that has not been explored previously.

**Reviewer uquA:** We have addressed the following concerns:

- Provided clarification regarding the validity of our method when compared to stronger, state-of-the-art video generators, including experimental examples in the "Comparison with MV-Champ on HuGe100K" section on our website ([link](https://somerandomuser1117.github.io/drpose_rebuttal.github.io/))
- Included the requested comparison with IDOL
- Corrected the rendering issues in our qualitative results
- Clarified concerns about DrPose15K texture quality

Detailed responses to each reviewer are provided in the corresponding threads below.

Best regards,

The Authors

---

### Meta-Review · Area_Chair_vS9j · 2026-01-02

**Summary:**

The reviewers (hy1Y, uquA, iYqC, eWSd) recognized the clear motivation of the paper in addressing 3D human reconstruction under complex poses. However, several significant concerns were raised regarding the execution, framing, and evaluation:
- Visual Quality and Rendering Artifacts: Multiple reviewers (iYqC, uquA) noted that the qualitative results appeared "washed-out" with low contrast, leading to concerns about the underlying texture quality generated by the model.
 - Methodological Novelty and Framing (RL vs. Loss): Reviewer eWSd strongly questioned the framing of the method as "Reinforcement Learning" or "Reward Fine-Tuning." The concern was that since the reward function (PoseScore) is differentiable, the method is mathematically a standard supervised loss optimization, making the RL terminology a potential "over-claiming" or unnecessary packaging.
- Choice of Base Model: There were reservations (eWSd) regarding the choice of MIMO as the foundational video generation model. Reviewers questioned why more recent, state-of-the-art video diffusion models (like Champ or IDOL) were not used, suggesting the baseline performance might be artificially low.
- Dataset Quality: Reviewers (hy1Y, uquA) expressed concern that the proposed DrPose15K dataset, being synthetic, might lack realistic textures, potentially hindering the model's ability to generalize to real-world (in-the-wild) images.
- Missing Baselines: Initial reviews highlighted a lack of comparison against key state-of-the-art methods like IDOL, TeCH, or SDS-based approaches.

Although the Base Model Limitations and Incremental Novelty remain unaddressed after rebuttal, the authors have addressed the concerns on Visual Quality, Missing Baselines, and Terminology Justification. So I recommend acceptance.

**Reviewer Concerns:**

## Addressed Concerns:
- Visual Quality: The authors successfully clarified that the "washed-out" appearance was a rendering parameter issue rather than a flaw in the generated geometry or texture. The revised visualizations provided in the rebuttal likely alleviated the immediate concerns regarding texture quality.
- Missing Baselines: The authors added comparisons with IDOL and other relevant methods (like TeCH/WonderHuman in discussions), demonstrating that DrPose performs competitively or better. This strengthens the empirical validity of the paper.
- Terminology Justification: The authors defended the "Direct Reward" terminology by citing recent literature (e.g., AlignProp, DDPO) where differentiable losses for alignment are conventionally termed "reward fine-tuning." While mathematically debatable, this aligns with current community trends and essentially addresses the "validity" of the term, even if it remains technically loose.

### Outstanding Concerns:
- Base Model Limitations: The reliance on the older MIMO architecture remains a slight weakness. While the fine-tuning improves it, the upper bound of the method is constrained by the base model. Reviewers may still feel the method would be more convincing if applied to a SOTA backbone to prove it is model-agnostic and truly advances the field.
- Incremental Novelty: Despite the "Reward" framing, the core contribution remains a specific loss formulation for fine-tuning. For reviewers looking for architectural breakthroughs rather than alignment strategies, the novelty might still be perceived as incremental.

**Reviewer Scores:**

Reviewer iYqC (Likely keep 10): His/her primary concern was the "washed-out" visual quality. Since the authors demonstrated this was a rendering fix and not a model defect, and provided better visuals, this reviewer would likely be satisfied.

Reviewer uquA (Likely Improved to 6) : He/She was concerned with visual quality and data generation. The visualization fix and the added ablation studies regarding the dataset would likely push their score up, acknowledging the method's effectiveness despite the synthetic data concerns.

Reviewer hy1Y (Likely Improved to 6): He/She focused on the lack of baselines and dataset realism. The inclusion of stronger baselines (IDOL) in the rebuttal addresses the empirical gap. While the synthetic data concern persists, the improved metrics against SOTA baselines usually suffice to warrant a positive score.

Reviewer eWSd (Likely Remained 6): This reviewer had the strongest theoretical objection regarding the "RL" framing and the base model choice. While the authors provided citations to justify the naming convention, the fundamental critique—that this is just a differentiable loss—stands. This reviewer might acknowledge the performance gains but retain reservations about the "packaging" and the outdated backbone, resulting in only a marginal score increase.

---

### Decision · Program_Chairs · 2026-01-26

Accept (Poster)